# The Relationship between *S. aureus* and Branched-Chain Amino Acids Content in Composite Cow Milk

**DOI:** 10.3390/ani9110981

**Published:** 2019-11-16

**Authors:** L. Grispoldi, M. Karama, F. Ianni, A. La Mantia, L. Pucciarini, E. Camaioni, R. Sardella, P. Sechi, B. Natalini, B. T. Cenci-Goga

**Affiliations:** 1Department of Veterinary Medicine, University of Perugia, Via San Costanzo 4, 06126 Perugia, Italy; grisluca@outlook.it (L.G.); pa.1981@hotmail.it (P.S.); 2Faculty of Veterinary Science, Department of Paraclinical Sciences, University of Pretoria, Onderstepoort 0110, South Africa; Musafiri.Karama@up.ac.za; 3Department of Pharmaceutical Sciences, University of Perugia, Via Fabretti, 48, 06123 Perugia, Italy; federica.ianni@chimfarm.unipg.it (F.I.); lucia.pucciarini@hotmail.it (L.P.); emidio.camaioni@unipg.it (E.C.); benedetto.natalini@unipg.it (B.N.); 4School of Advanced Studies, University of Camerino, Via Camillo Lili 55, 62032 Camerino, Italy; alessandro.lamantia@unicam.it

**Keywords:** branched-chain amino acid, ion-pair reversed-phase liquid chromatography, mastitis, dairy cow, *Staphylococcus aureus*

## Abstract

**Simple Summary:**

*Staphylococcus aureus* is not only a common cause of bovine mastitis, but also an agent of food poisoning in humans. *S. aureus* has the ability to produce branched-chain amino acids (BCAAs) under certain nutritional conditions. We show that levels of two BCCA (leucine and isoleucine) are correlated to the load of *S. aureus* in composite milk samples. The application of the ANOVA and Tukey-Kramer analysis showed statistically significant differences in the content of the BCAAs, isoleucine and leucine, between the two groups based on *S. aureus* positivity (*p* < 0.001), while bivariate Pearson correlation analysis showed a strong relationship between *S. aureus* load and the content of these two BCAAs.

**Abstract:**

The early diagnosis of mastitis is an essential factor for the prompt detection of the animal for further actions. In fact, if not culled, infected cows must be segregated from the milking herd and milked last, or milked with separate milking units. Besides microbiological analysis, the somatic cell count (SCC) commonly used as predictor of intramammary infection, frequently lead to a misclassification of milk samples. To overcome these limitations, more specific biomarkers are continuously evaluated. The total amino acid content increases significantly in mastitic milk compared to normal milk. *S. aureus* requires branched-chain amino acids (BCAAs—isoleucine, leucine, and valine) for protein synthesis, branched-chain fatty acids synthesis, and environmental adaptation by responding to their availability via transcriptional regulators. The increase of BCAAs in composite milk has been postulated to be linked to mammary infection by *S. aureus*. The aim of this work is to demonstrate, by a direct ion-pairing reversed-phase method, based on the use of the evaporative light-scattering detector (IP-RP-HPLC-ELSD), applied to 65 composite cow milk samples, a correlation between the concentration of isoleucine and leucine, and *S. aureus* load. The correlation coefficient, r, was found to be 0.102 for SCC (*p* = 0.096), 0.622 for isoleucine (*p* < 0.0001), 0.586 for leucine (*p* < 0.0001), 0.013 for valine (*p* = 0.381), and 0.07 for tyrosine (*p* = 0.034), standing for a positive correlation between *S. aureus* and isoleucine and leucine concentration. The link between the content of BCAAs, isoleucine and leucine, and udder infection by *S. aureus* demonstrated with our study has an important clinical value for the rapid diagnosis of *S. aureus* mastitis in cows.

## 1. Introduction

*Staphylococcus aureus* causes one of the most common types of chronic mastitis in cows. Though some cows may show signs of clinical mastitis (especially after calving), the infection is usually subclinical, often with no detectable changes in milk or the udder. *S. aureus* have the ability to survive and multiply in the mammary gland tissues and are contagious. When the infection is established is hard to treat with antimicrobial therapy and the infected subjects must be segregated from the rest of the herd in order to avoid the spread of the bacteria that usually occurs at milking time [1]. It is well known that it is hard to eradicate the presence of *S. aureus* by using standard milking-time hygiene techniques in herds with low levels of SCC (<200,000 cells/mL) [2,3,4]. Schukken, et al. [5] have shown that *S. aureus* can reach a prevalence of 3% in all producing dairy cows; and it was observed that it can cause 10 to 12% of clinical forms (the values are variable from farm to farm) [6]. There is evidence that between 12 and 15% of heifers are infected with *S. aureus* after the first calving [7,8,9].

According to the EU Regulation (Regulation (EC) No 853/2004 of the European parliament and of the Council of 29 April 2004 laying down specific hygiene rules for food of animal origin) raw milk must meet the following criteria—plate count at 30 °C ≤ 100,000 colony forming unit (cfu)/mL and somatic cell count ≤ 400,000 cells/mL. However, since several factors (i.e., animal age, breed, stage of lactation, the time of the day at sampling, and the daily frequency of milking) can influence this value. Some perplexities are correlated to the exclusive use of SCC to diagnose subclinical infection.

Interestingly, cows infected with *S. aureus* do not necessarily have elevated SCC. Jones et al. back in 1984 [10] demonstrated that only 60% of cows infected with *S. aureus* produced milk with more than 300,000 somatic cells/mL.

In this scenario, the early diagnosis of mastitis, especially at subclinical stages, becomes a pivotal factor for a prompt consideration and detection of the animal, thus minimizing the related consequences in terms of transmission of infection.

So far, several methods have been developed for the rapid diagnosis of subclinical mastitis. The SCC in milk, a marker of cells present in milk including inflammatory cells, is an indicator of intramammary inflammation and consequently a predictor of intramammary infection. An elevated SCC in raw milk has a negative influence on its quality.

The results of many studies [11,12,13] suggest that cows with SCC levels lower than 200,000 cells/mL have not been infected with major mastitis pathogens (namely, *S. aureus* and Streptococcus agalactiae), while cows with a SCC above 300,000 cells/mL are probably infected. Moreover, the use of quarter milk samples for routine udder health monitoring has become expensive for large dairy herds, and initial udder health surveys are now conducted using composite cow milk samples [14]. Petzer et al. [14] in 2017 conducted a thorough study on the validity of somatic cell count as indicator of pathogen-specific intramammary infection which revealed that at a 200,000 cells/mL SCC threshold, the sensitivity for detecting major Gram-positive, major Gram-negative, and minor pathogens was 79.9%, 95.5%, and 51.7%, respectively, with a specificity of 50.3%, 52.8%, and 53.5%, respectively. At a 150,000 cells/mL SCC threshold, the sensitivity improved to 84.2%, 96.1%, and 60.1%, respectively, for the same microbial groups.

In order to overcome such limitations, efforts have been spent by scientists in search of suitable biomarkers for the rapid diagnosis of subclinical stages of cow mastitis [15,16,17]. In previous studies [18,19,20,21] it was found that the total amino acid content increased significantly in mastitic milk compared to milk from healthy udders. In particular, the levels of branched-chain amino acids (BCAAs) and some aromatic amino acids (AAAs) were found to be altered at subclinical stages. Moreover, *S. aureus* requires branched-chain amino acids (BCAAs—isoleucine, leucine, and valine) for protein synthesis, branched-chain fatty acids synthesis, and environmental adaptation. *S. aureus* needs to either synthesize BCAAs or scavenge them from the environment and it responds to their availability via transcriptional regulators. The increase in BCAAs in composite milk has been postulated to be linked to mammary infection by *S. aureus* [22,23,24,25]. Despite encoding the BCAA biosynthetic operon, *S. aureus* relies on the acquisition of BCAAs, most importantly leucine and valine, for rapid growth in media with excess or limiting concentrations of BCAAs, indicating that BCAA biosynthesis is typically repressed [26,27]. Paradoxically, biosynthesis remains repressed even in the absence of an exogenous source of leucine or valine, with the growth of *S. aureus* observed only after a prolonged period, likely explaining why previous studies have been misled to conclude that *S. aureus* is auxotrophic for leucine and valine [24,28].

In an attempt to correlate the milk content of the BCAAs isoleucine, leucine and valine, and the aromatic amino acid, tyrosine, to *S. aureus* load, a new diagnostic tool based on direct HPLC analysis by ion-pairing reversed-phase (IP-RP-HPLC-ELSD) was developed and validated with 65 composite milk analyzed for SCC and *S. aureus* content.

## 2. Materials and Methods

### 2.1. Milk Sampling

The analysis was performed on 65 samples of milk collected at a dairy farm with known *S*. *aureus* problems. The dairy farm of choice was composed by 300 lactating Holstein Friesian cows and was located in the Umbria region, central Italy. The average milk production was 30.2 Kg/day, all sampled cows were in their second or third lactation and between week-10 and week-12 of lactation. The samples from the composite quarter collection of individual cows were transported to the laboratory for SSC measurement, *S*. *aureus* isolation and identification, and then frozen at −80 °C prior to further analysis.

The sample size was calculated using the formula:n = Z2 * p * (1−p)/C2,
where Z is the Z-value (e.g., 1.96 for a 95% confidence level), p is the expected prevalence, expressed as a decimal, and C is the confidence interval, expressed as a decimal [29]. With an expected prevalence of 50% (0.5), a confidence interval of 10, and a confidence level of 0.95, a sample size of 63 animals is then required. With 300 lactating cows in the farm, the sample size provides a 95% confidence level (CL) for cow-level prevalence, with a confidence interval (CI) of 10.

### 2.2. Somatic Cells Count

SCCs were measured at the APA (associazione provinciale allevatori) laboratory using a DeLaval cell counter according to the manufacturer’s instructions (Cell counter DCC; DeLaval, Tumba, Sweden). 60 µL of sample were aspirated into a small cassette that contained a DNA-specific fluorescent reagent that was bound to the SCC nuclei. The machine counted the fluorescent SCC nuclei in milk using an integrated digital camera.

### 2.3. S. Aureus Identification

The culture and identification of *S. aureus* isolates was done according to the methods described by Cenci Goga et al. [30]. Briefly, the first isolation medium was tryptose blood agar containing washed bovine red blood cells on which 1 mL aliquot of milk was spread and incubated at 37 °C for 48 h. Creamy, grayish-white or golden-yellow colonies, 3 to 5 mm in diameter, with a distinct zone of hemolysis, were considered to be presumptive *S. aureus*. These colonies were selected and tested for the following characteristics—cell morphology after Gram staining, production of catalase and coagulase, and for thermonuclease determination (TNase). Coagulase determination was performed using lyophilized rabbit plasma with EDTA (BBL Microbiology Systems, Cockeysville, MD) reconstituted with sterile water. Suspect *Staphylococcus* spp. colonies were inoculated into 5 mL test tubes containing 0.5 mL of the reconstituted plasma. Clotting after incubation at 37 °C between 4–24 h was regarded as a positive result. TNase determination was performed according to the method described by Ibrahim [31]. After an 18-h incubation at 37 °C on Brain Heart Infusion (BHI) broth (Difco, BD Diagnostic Systems, MD), the cultures were put into a water bath at 100 °C for 15 min to eliminate any non-specific heat-labile nuclease activity, and centrifuged at room temperature for 30 min at 3000 rpm before testing. One hour before testing, 5 mm wells were made on plates containing Thermonuclease Agar with Toluidine Blue (Remel, Lenexa, KS). Approximately 70 µL of supernatant was then transferred into each well and the plates were incubated at 37 °C for 4 h. The presence of a pink halo surrounding the wells was regarded as a positive result. All Gram-positive cocci, catalase and coagulase positive, and TNase-producing organisms were identified as *S. aureus*.

### 2.4. Bacterial Strain and Artificial Specimen Preparation for Baseline Contamination

In order to quantify the *S*. *aureus* load from the 65 frozen milk samples, a procedure based on milk dilution followed by DNA extraction and nested PCR was adopted.

*S. aureus* ATCC 29213 was used to set up a baseline contamination level and adulterated milk specimens were prepared as follows. A *S. aureus* culture was grown in Nutrient Broth (NB, Oxoid CM0001, Basingstoke, UK) at 37 °C for 48 h on air. The total viable cells (TVC) count (on Nutrient Agar, NA, Oxoid CM0003, incubated at 37 °C on air for 24 hrs) at 24 h was 1 × 10^8^ CFU/mL. A 10 mL aliquot of the culture was then added to 90 mL of raw milk to obtain a final concentration of 10^7^ CFU/mL. Decimal dilutions were performed to obtain the following concentration in milk—10^6^ CFU/mL, 10^5^ CFU/mL, 10^4^ CFU/mL, 10^3^ CFU/mL, 10^2^ CFU/mL, and 10 CFU/mL. The TVC counts from all samples were recorded, as a control, on NA. Raw milk used had been previously tested for the absence of contaminating *Staphylococcus* spp.

### 2.5. DNA Extraction

Sixty-five milk samples, after DNA extraction, were tested by two-step PCR for the detection of *S. aureus.* The milk samples (1 mL), from both the real study cases and the previously prepared standard samples, containing *S. aureus* (0–10^6^ CFU/mL) were each mixed with 1 mL phosphate-buffered saline (0.05% (v/v) Tween-20; PBST), vortexed, and centrifuged at 10,000 *g* for 5 min. This procedure was performed to pellet the bacterial cells and to remove the proteins and lipids in these milk samples, which may interfere with PCR amplification. Following the removal of the supernatant fluid, the DNA was extracted using a sodium iodide (NaI) method [32]. The NaI extraction procedure was a modification of the method proposed by Ishizawa [33]. The pellet was re-suspended in 1 mL of TE buffer (10 mM Tris HCl pH 8.0, 1 mM EDTA pH 8.0), transferred to a 2 mL microcentrifuge tube and added with 200 µL ammonium hydroxide, 200 µL absolute ethanol, 400 µL petroleum ether, and 20 µL sodium dodecyl sulfate (SDS) 10%. The mixture was vortexed and then centrifuged at 15,000 *g* for 10 min. After supernatant removal, 1 mL of PBST was added and 100 µL transferred to a 2 mL microcentrifuge tube. A 300 µL volume of a solution containing 6 M NaI/13 mM EDTA/0.5% sodium N-lauroylsarcosine/10 µg glycogen/26 mM Tris-HCl, pH 8.0 were added to the tube, mixed, and incubated at 60 °C for 15 min in a heating block. After addition of an equal volume of isopropanol, the mixture was vigorously agitated, and let stand for 15 min. The sample was then centrifuged at 10,000 *g* for 5 min to precipitate the DNA and the supernatant was discarded. A 1 mL volume isopropanol 40% was added and the mixture was vortexed. After centrifugation at 10,000 *g* for 5 min to recover the DNA, the pellet was vacuum-dried. All operations were conducted at room temperature (RT). The extracted DNA was stored at −20 °C prior to PCR.

### 2.6. Oligonucleotide Primers and DNA Amplification

All samples were tested to detect the presence of the *nuc* gene by means of PCR. To obtain very sensitive detection, a two-step PCR amplification procedure was developed with two nested sets of primers. In the first PCR, the primers, *nuc_ext*-f (5′ GCGATTGATGGTGATACGGTT 3′) and *nuc_ext_int*-r (5′ GCCAAGCCTTGACGAACTAAAGC 3′), were used (MWG Biotech, Ebersberg, DE). The primers, *nuc_e*-f and *nuc_ext_int*-r, are located at positions 384 to 404 and 639 to 661, respectively, of the coding sequence (accession number NZ_LHUS02000183.1). By means of these primers, a fragment of 278 base pairs (bp) was amplified. For the nested PCR, the primer, *nuc_i*-f (5′ AAAATGCAAAGAAAATTGAAGTC 3′), was used in combination with the same primer, *nuc_ext_int*-r, of the first PCR (5′ GCCAAGCCTTGACGAACTAAAGC 3′) (MWG Biotech). They are located at positions 152 to 174 and 256 to 278, respectively, of the amplicon obtained after the first PCR. The nested PCR amplified a DNA fragment of 127 bp. A 5 μL volume of each extracted sample was used for PCR, which also contained 1.5 mM MgCl_2_, 0.2 mM of each dNTP, 2.5 U of Taq DNA polymerase (Taq DNA Polymerase in Storage Buffer A, M1865, Promega Corporation, Madison, WI, USA), and 0.1 mM of each appropriate primer, in a total volume of 25 µL. For the first PCR, 5 µL of the sample DNA was added; for the second PCR, 5 µL of the first PCR product was used as template. The DNA amplification reactions were carried out using a PCR Mastercycler (Eppendorf AG, Hamburg, DE) with the following programme: first PCR—denaturation at 95 °C for 3 min, 30 cycles each consisting of 1 min denaturation at 95 °C, 1 min annealing at 65 °C, 1 min extension at 72 °C, and a final extension for 10 min at 72 °C; nested PCR—denaturation at 95 °C for 3 min, 30 cycles each consisting of 1 min denaturation at 95 °C, 1 min annealing at 54 °C, 1 min extension at 72 °C, and a final extension for 10 min at 72 °C. In each PCR assay, a positive control with 100 ng of *S. aureus* ATCC 29213 DNA and a negative control without any bacterial DNA were included. A 10 µL aliquot of each PCR product was subjected to 1% (w/v) agarose gel electrophoresis, containing 0.5 µg/mL ethidium bromide (Promega Corporation, M 5041), for 30 min at 100 V. The PCR products (278 bp and 127 bp for the first PCR and the nested-PCR, respectively) were visualized under UV illumination. Their size was estimated using a standard DNA molecular weight marker (Novagen 69278-3, Madison, WI, USA).

For the 65 samples, 10-fold dilutions of milk were performed prior to DNA extraction and the PCR results were recorded as the highest dilution still positive at the nested PCR. For the “artificial specimen preparation”, *S. aureus* ATCC 29213 was cultured in NB and added to raw milk. Following plating of each dilution on NA, the numbers of CFU/mL were in accordance with the expected values based on the decimal dilution factor. No *S. aureus* DNA was detectable in the unadulterated milk. Bacterial DNA was successfully extracted from adulterated raw milk by the NaI method. After the first PCR, the NaI method provided an upper detection limit of 10^6^ CFU/mL but not a satisfying lowest limit. The second PCR with nested primers allowed a sensitive improvement in terms of detection, thus allowing detection of a lowest limit of 10 CFU/mL. No visible amplicons were obtained from control samples where no *S. aureus* had been added.

### 2.7. HPLC Measurements and Extraction of Free Amino Acids from the Raw Milk Sample

Free amino acidic forms were extracted from raw milk samples, according to the procedure described elsewhere [21]. Before the analysis with the previously validated IP-RP-HPLC-ELSD method, the extracts were lyophilized and then examined individually at a 20 mg/mL concentration. For the chromatographic assay, each extract was dissolved in a hydro-alcoholic solution, and then eluted under the gradient conditions described by Ianni et al. [21].

The HPLC analyses were performed on a Shimadzu (Kyoto, Japan) LC-20A Prominence, equipped with a CBM-20A communication bus module, two LC-20AD dual piston pumps, an SPD-M20A photodiode array detector, and a Rheodyne 7725i injector (Rheodyne Inc., Cotati, CA, USA) with a 20 μL stainless steel loop. A Varian 385-LC evaporative light scattering detector (ELSD) (Agilent Technologies, Santa Clara, CA, USA) was used to follow the HPLC runs. The analog-to-digital conversion of the output signal from the ELSD was obtained by using a common interface device. The applied ELSD setting for the analysis was: 30 °C nebulisation temperature, 50 °C evaporation temperature, 1 L/min gas flow rate (air) and 1 as the gain factor.

A Prevail RP18 column (Grace, Sedriano, Italy) (250 mm × 4.6 mm i.d., 5 μm, 110Å pore size), was used as the analytical column for the HPLC analysis.

The details on the procedure applied to extract the free amino acidic content from raw milk samples have been reported elsewhere along with the experimental conditions for the IP-RP-HPLC-ELSD analyses [21]. By relying upon the heptafluorobutiryc acid (HFBA) as the ion-pairing (IP) reagent [34,35], the developed chromatographic method revealed to be suitable for an efficient direct analysis of unbound proteinogenic amino acids in raw milk. In particular, the method was highly selective for hydrophobic amino acids, including BCAAs and AAAs, which were base line resolved from each other and from other matrix interferences. As a consequence, a trustful quantification of these compounds was made possible.

Direct methods avoid pre-analysis chemical derivatization steps of the compounds under investigation, thus overcoming the drawbacks intrinsically related to indirect analyses—(i) the production of interfering by-products due to the presence of the derivatizing reagent; (ii) the possible different derivatization rate for the different amino acids under investigation; (iii) the non-quantitative yield of the derivatization reaction and/or the non-quantitative recovery by the following purification step. All these aspects can cumulatively lead to a misquantification of the amino acidic content in the investigated samples. In this setting, the use of HFBA as the IP-reagent offers the additional advantage of increased residence times of polar analytes into the RP system, thus improving the overall chromatographic performance quality. Nonetheless, unlike the use of other perfluoroalkyl carboxylic acids as ion-pairing reagents, the use of HFBA avoids the prolonged re-equilibration times of the chromatographic systems between consecutive analyses. Still, the high volatility of this IP-agent, makes it compatible with mass spectromety (MS) detectors, thus revealing the whole applied method suitable for accurate molecular investigations [36,37].

### 2.8. Statistical Methods

Statistical analyses were performed with the aid of StatView for Mac OS (SAS Institute, Inc. Cary, NC, USA). Pearson correlation and one-way ANOVA (Analysis of Variance) were used as a statistical test to assess the differences in means between the groups. In order to find a relationship between the *S*. *aureus* load associated with each milk sample and the SCC and the concentration of Ile, Leu, Val, and Tyr, a bivariate Pearson correlation analysis was conducted. Tukey-Kramer test at a confidence level of 95%, was further employed for multiple comparisons between all pair-wise means to determine how they differ. A *p* < 0.05 was considered statistically significant.

For sensitivity and specificity, the values were calculated with the results from PCR as true positive and true negative and using the following formulas:sensitivity = a / (a + c)(1)
specificity = b / (b + d)(2)
where a is number of true positives, b is number of true negatives, c is number of false negatives, and d is number of false positives.

## 3. Results and Discussion

The results are summarized in Table 1. The prevalence of *S. aureus* ranged from 40.74% in milk samples with SCC < 100,000 to 87.5% in milk samples with SCC between 200,000 and 400,000. The isoleucine content ranged from 111 to 168 µg/mL, leucine from 126 to 250, valine from 226 to 428, and tyrosine from 97 to 127. The *S*. *aureus* load ranged from 0.89 log cfu/mL (sd 1.37) in milk samples with SCC < 100,000 to 2.80 log cfu/mL (sd 1.64) in milk samples with SCC > 1,000,000. The relationship between the *S*. *aureus* load associated with each milk sample and the SCC and the concentration of Ile, Leu, Val, and Tyr, determined by a bivariate Pearson correlation analysis, is shown in Figure 1, Figure 2, Figure 3, Figure 4 and Figure 5. The correlation coefficient, *r*, was found to be 0.102 for SCC (*p* = 0.096), 0.622 for isoleucine (*p* < 0.0001), 0.586 for leucine (*p* < 0.0001), 0.013 for valine (*p* = 0.382), and 0.07 for tyrosine (*p* = 0.034), standing for a positive correlation between *S*. *aureus* and isoleucine and leucine concentration. No correlation was observed between *S*. *aureus* and SCC, and between *S*. *aureus* and valine and tyrosine. Bar plot analysis (ANOVA and Tukey-Kramer test) (Figure 6 and Figure 7) confirmed the differences in the amino acid content between the two groups of milk samples (*S*. *aureus* positive and *S*. *aureus* negative). This represents the first study in which a direct chromatographic method has revealed a statistically significant difference (*p* < 0.0001) in the content of isoleucine and leucine between *S*. *aureus* positive and *S*. *aureus* negative composite milk samples. On the other hand, the content of valine and tyrosine was not statistically different in the two groups (*p* = 0.761 for valine and *p* = 0.354 for tyrosine). These findings (no correlation between SCC and *S*. *aureus* load and strong correlation between *S*. *aureus* and isoleucine and leucine concentration) bring into question the use of the SCC test as a survey tool to identify intramammary infections caused by *S*. *aureus* under field conditions, especially for an eradication campaign. In fact, although numerous factors can influence the SCC at the individual cow- and udder-quarter level, such as parity, lactation stage, incorrect milking machine settings, stress, and other factors including genetics, the most important cause remains the infection status of the mammary gland [14]. Table 2 show the sensitivity (*se*) and specificity (*sp*) calculated for the SCC and for the isoleucine and leucine content used to screen for *S*. *aureus* positive composite milk samples. From these data, along with data from Table 1, it appears that it is important to know how reliable SCCs of composite milk samples are as indicators of intramammary infection and *S*. *aureus* culture-positive results. Our data (Table 2) suggest that the best combination for *se* and *sp* is for a SCC threshold of 150,000 cells/mL (se 0.58; 0.41–0.75, sp 0.67; 0.51–0.83) and that the highest *se* is obtained when the threshold is lowered to 100,000 cells/mL (0.71; 0.57–0.85). High *se* is needed when surveying a herd for the purpose of identifying samples with positive bacterial growth. Our data (Table 2) prove also that, in the herd studied, the isoleucine and leucine content, at a threshold of 100 µg/mL, have always had higher *se* and *sp* values than SCC. These results, in combination with data from the previous studies of our research group [20,21], indicate that the content of the BCAAs, isoleucine and leucine, are a promising marker of udder infection by *S*. *aureus*. Multiplication of this bacterium within the mammary gland is required for infection to persist and in recent years the nutrient requirements of *S*. *aureus* have been studied and how these relate to the availability of nutrients in vivo has been determined. The branched-chain amino acids (BCAAs) are vital to both the growth and virulence of *S*. *aureus*. In addition to supporting protein synthesis, the BCAAs serve as precursors for branched-chain fatty acids (BCFAs), which are predominant membrane fatty acids, and in association with the global regulatory protein CodY, the BCAAs are key co-regulators of virulence factors [27]. Moreover, it has been demonstrated that it is not completely true that *S*. *aureus* is auxotrophic for leucine and valine, in fact it carries an intact BCAA biosynthetic operon and produces leucine and valine, although only under certain nutritional conditions. Isoleucine levels also influence expression of genes involved in the ability of *S*. *aureus* to cause disease [23]. The link between BCAAs isoleucine and leucine content and udder infection by *S*. *aureus* demonstrated with our study, along with findings from other authors regarding its amino acid requirements [23,24,25,26,27,38] paired with the direct chromatographic method proposed by our research group [20,21], has an important clinical value for a rapid diagnosis of *S*. *aureus* mastitis in cows.

## 4. Conclusions

*S*. *aureus* mastitis comes with widespread metabolic perturbations and the BCAAs appear to be among the most distinctly perturbed metabolites. In the present work, an ion-pairing reversed-phase method, based on the use of the evaporative light-scattering detector and on the use of the heptafluorobutyric acid as the ion-paring agent, was successfully applied for the direct analysis of the amino acid content in 65 milk samples. The application of the ANOVA and Tukey-Kramer analysis shed light on statistically significant differences in the content of the BCAAs, isoleucine and leucine, between the two groups based on *S*. *aureus* positivity (*p* < 0.001), while the bivariate Pearson correlation analysis showed a strong relationship between *S*. *aureus* load and the content of these two BCAAs. Therefore, a great sensibility of isoleucine (Ile) and Leucine (Leu) was observed in this study suggesting that they are suitable as biomarkers for the diagnosis of subclinical mastitis caused by *S*. *aureus*. Grounded on these evidences, it is clear that there could be a pronounced deal of clinical value in monitoring of isoleucine and leucine levels in milk, given also the low correlation between SCC and *S*. *aureus*.

## Figures and Tables

**Figure 1 animals-09-00981-f001:**
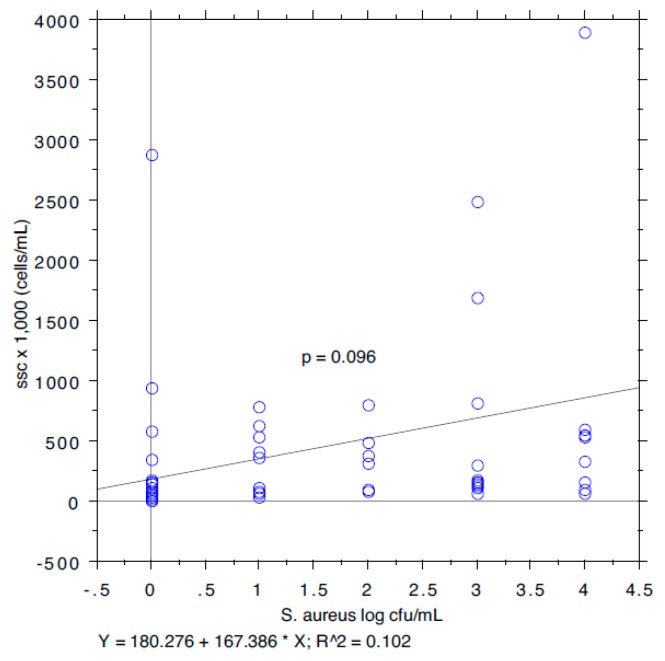
Pearson correlation between SCC and *S*. *aureus* load.

**Figure 2 animals-09-00981-f002:**
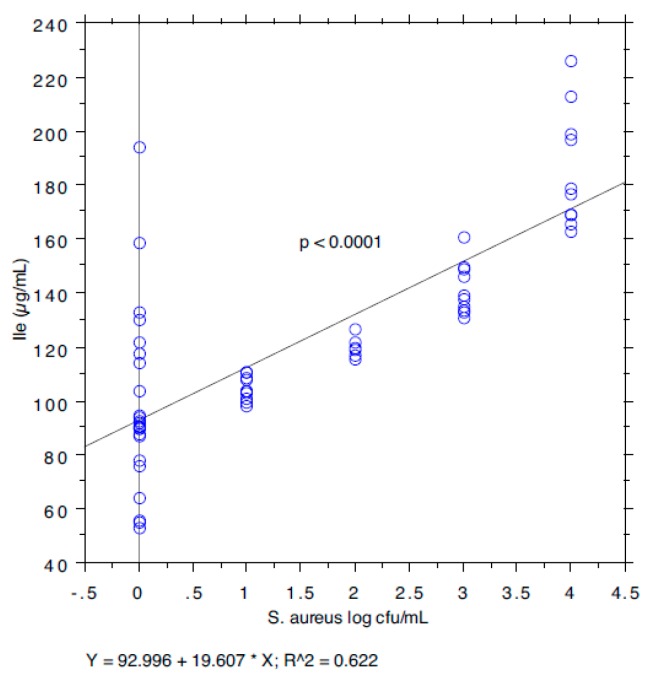
Pearson correlation between isoleucine content and *S*. *aureus* load.

**Figure 3 animals-09-00981-f003:**
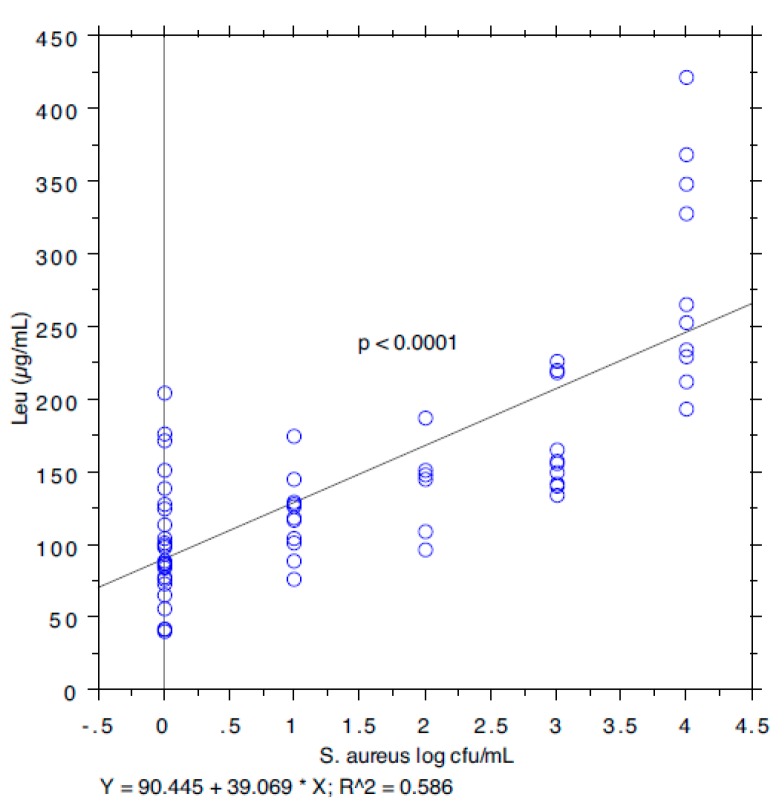
Pearson correlation between leucine content and *S*. *aureus* load.

**Figure 4 animals-09-00981-f004:**
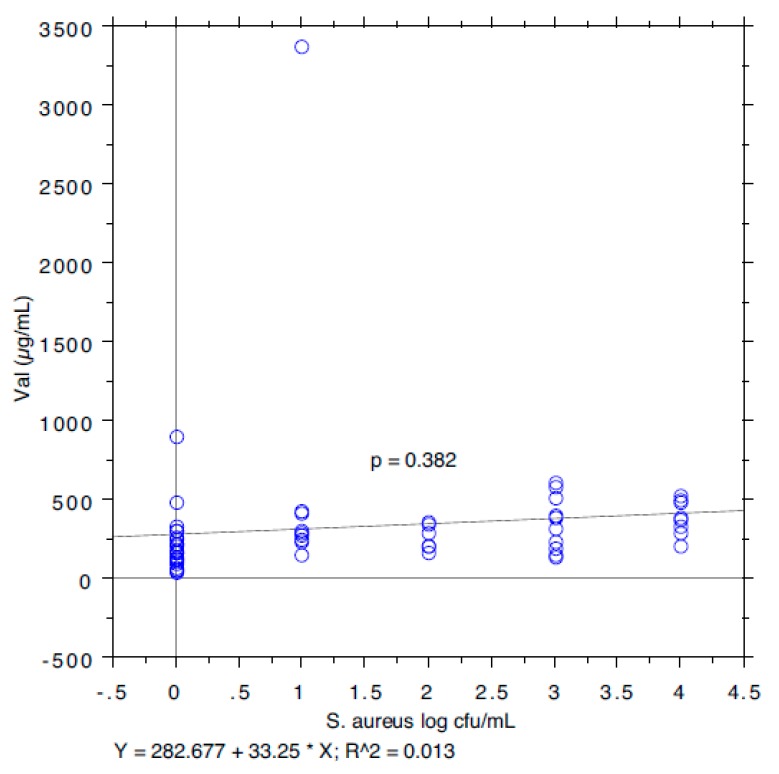
Pearson correlation between valine content and *S*. *aureus* load.

**Figure 5 animals-09-00981-f005:**
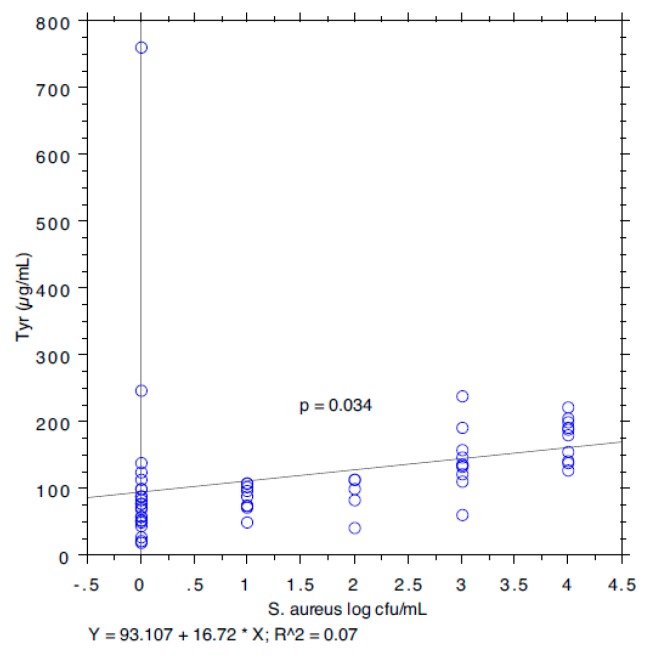
Pearson correlation between tyrosine content and *S*. *aureus* load.

**Figure 6 animals-09-00981-f006:**
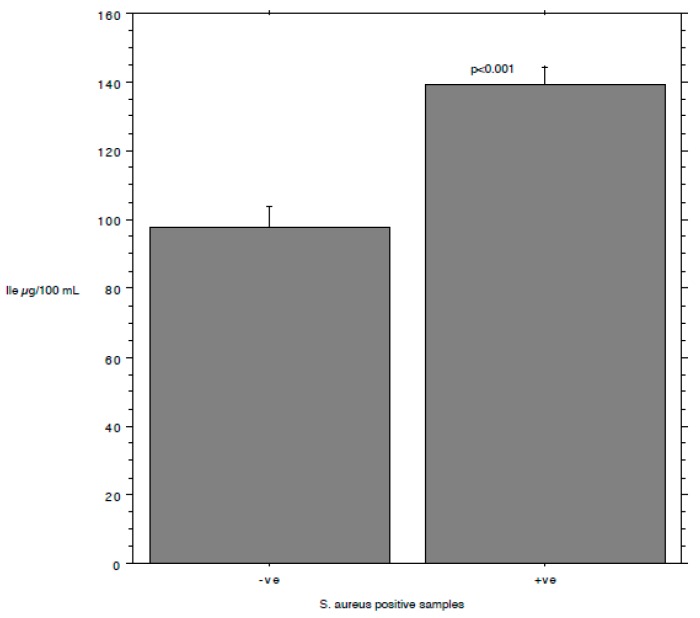
Bar plot for isoleucine content (mean and standard error) and ANOVA/Tukey-Kramer test comparing *S*. *aureus* positive and negative samples.

**Figure 7 animals-09-00981-f007:**
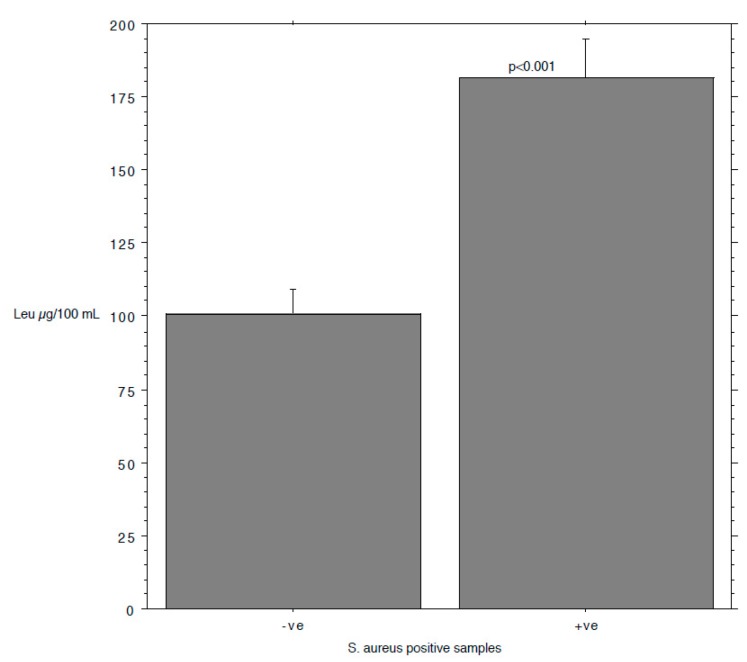
Bar plot for leucine content (mean and standard error) and ANOVA/Tukey-Kramer test comparing *S*. *aureus* positive and negative samples.

**Table 1 animals-09-00981-t001:** SCC, *S*. *aureus* load, and selected amino acids content for 65 composite milk samples.

	ssc Count (× 1000)		*S*. *aureus* + ve (n)	*S*. *aureus* Load (log cfu/mL)		Ile (µg/mL)		Leu (µg/mL)		Val (µg/mL)		Tyr (µg/mL)	
Scc Groups	Mean	sd		Mean	sd	Mean	sd	Mean	sd	Mean	sd	Mean	sd
< 100 (n = 27)	43.6	32.6	11	0.89	1.37	118	32	135	57	428	638	127	138
100–199 (n = 15)	142.6	26.1	8	1.53	1,60	111	33	126	51	256	153	97	60
200–499 (n = 8)	364.3	57.8	7	1.88	1,25	118	34	144	49	246	139	113	51
500–999 (n = 10)	672.2	147.5	8	2.00	1,63	128	47	173	109	226	104	109	43
> 1000 (n = 5)	2967.2	941.3	4	2.80	1,64	168	54	250	143	335	203	165	58
All (n = 65)	427.5	808.9	38	1.48	1,54	122	38	149	79	330	433	118	97

**Table 2 animals-09-00981-t002:** Sensitivity (se) and specificity (sp) with 95% lower (LCI) and upper (UCI) confidence intervals for various SCC, isoleucine (Ile), and Leucine (Leu) threshold.

Threshold	se	95% LCI and UCI	sp	95% LCI and UCI
SCC > 400,000/mL	0.37	(0.14–0.60)	0.89	(0.80–0.98)
SCC > 200,000/mL	0.50	(0.30–0.70)	0.85	(0.74–0.96)
SCC > 150,000/mL	0.58	(0.41–0.75)	0.67	(0.51–0.83)
SCC > 100,000/mL	0.71	(0.57–0.85)	0.59	(0.19–0.41)
Ile > 100 µg/mL	0.97	(0.93–1.00)	0.70	(0.50–0.90)
Ile > 150 µg/mL	0.32	(0.07–0.56)	0.93	(0.85–1.00)
Leu > 100 µg/mL	0.92	(0.84–1.00)	0.52	(0.28–0.76)
Leu > 100 µg/mL	0.55	(0.37–0.74)	0.78	(0.65–0.91)

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
