# Peer review of "The Relationship between S. aureus and Branched-Chain Amino Acids Content in Composite Cow Milk"

_animals, 2019, doi:10.3390/ani9110981_

Round 1

Reviewer 1 Report

the manuscript was improved and it is suitable for publication in the present form

line 252 change "Materials and Methods" with "Results and Discussions"

Author Response

We have amended the title.

Reviewer 2 Report

All suggestions are taken in considerations by authors.

Just to one correction:

L252: Please insert “3. Results and Discussion” instead of “3. Materials and Methods”.

And, one suggestion:

L313-314: This very low regression coefficient is not significant. So I suggest to remove the fig.4.

Author Response

L252: Please insert “3. Results and Discussion” instead of “3. Materials and Methods”.

Response:Corrected. thank you.

>L313-314: This very low regression coefficient is >not significant. So I suggest to remove the fig.4.

Response:We concur that fig. 4 shows no statistically significant correlation, however in the text we state that: «In order to find a relationship between S. aureus load associated to each milk sample and SCC and the concentration of Ile, Leu, Val and Tyr, ***a bivariate Pearson correlation analysis was conducted***.Tukey-Kramer test at confidence level of 95%, ***was further*** employed for multiple comparisons between all pair-wise means to determine how they differ. p< 0.05 was considered statistically significant».

We defer to reviewer and to editor, but we believe that the results of the first test (Pearson) should be left in place, since then we only show the significant associations.

This manuscript is a resubmission of an earlier submission. The following is a list of the peer review reports and author responses from that submission.

Round 1

Reviewer 1 Report

Dear Editor,

thank you for consider me for the revision of the manuscript animals-608170 entitled “The relationship between S. aureus and branched-chain amino acids content in composite cow milk”.

The manuscript is about the study of the content of branched-chain amino acids and their correlation with S. aureus in composite cow milk. The use of amino acids as biomarkers able to indicate S. aureus infection could be a promise strategy aimed to reduce the consequences of mastitis in cow, by an early diagnosis, especially in subclinical stages.

This is an interesting matter, and the results presented could be of inspiration to start with further investigations on the topic.

However, in my opinion the manuscript, in the present form is acceptable for a publication as research article only after major revision, for the reasons listed below.

The main concerns are:

Introduction. Even if very well written, in my opinion it’s too long. In particular the authors describe in a very detailed way (line 51 to 90) the use of SCC in the diagnosis of mastitis and all the related limitation, which is not the topic of the manuscript. I think this part should be shortened.

Results and Discussions. The first five lines must be moved in the M&M section. A part from the description of the results (reported in the tables), the comments and the discussions are quite poor. Only three citation are used to discuss data, two of that are of the group of research of the authors.

Excepted for the literature cited in the M&M section, almost all the references are used in the introduction; for a research article, more importance and emphasis should be given to comment the data, for the implication of the results etc., with the help of literature.

(line 252 254) I accept that, as the authors stated, the manuscript represents the first study … but for a research article a more deeply discussion of the data is request. This is of crucial importance, in my opinion for the acceptance of the manuscript.

Minor concerns

Line 159-160. Close the bracket.

Line 233. “P < 0.005” probably the authors mean 0.05

Lines 235 to 239. Move in the material and methods section.

Some of the references in the test are not reported in the numeric form:

 For example, Grispoldi et al., 2019 (line 46) and Jones, 1984 (line 57) are also not reported in the list of references).

Line 83 “Petzer, Karzis, Donkin, Webb and Etter” should be “Petzer et al.“

Line 136 Cenci Goga et al (2003) should be numeric

Line 239 “Ianni, Sechi, La Mantia, Pucciarini, Camaioni, Cenci Goga, Sardella and Natalini” should be “Ianni et al.”

Figures: the size of the figures could be reduced and grouped in 2 or 3 figures.

Author Response

Please find enclosed a pdf file with our answers

Reviewer 2 Report

General comments:

This study aimed to validate a new method using some potential BCCA as biomarkers and in consequence to predict subclinical intramammary infections by S. aureus in dairy cows.

The design of the study is adequate and the findings seems to be valid presenting the validation of the new method the main novelty, once the SCC presents some limitations.  Nevertheless, some part of the manuscript remains unclear or the content is not appropriate for the respective section and should be corrected. Also some statistical significance of correlations should be clarified. Once the main aim is to validate a new method for diagnosis purpose, we suggest to give more details regarding the obtention of Se and Sp and even add the negative and positive predictive values.

Specific comments:

L15-19: The simple summary need to give the main finding of the study. Please re-write it according.

L20-36: Also here authors need to rewrite the abstract according the journal rules reporting at least the main aim, results and conclusions. The style of the abstract is close to https://doi.org/10.3390/app9020349

L46: “…segregated… “ instead of “…divided…”. Also Change (Grispoldi et al., 2019) to a numerical reference. Also review whole text for citation format (there are several errors).

L48-50: please reformulate this sentence: you means that is observed that S. aureus can reach a prevalence of 3% in all producing dairy cows; and it was observed that can cause 10 to 12% of Clinical forms. (the values are variable from farm to farm).

L59: Probably, this sentence is outside of the context and should be inserted above in L50.

L63: “…subclinical S. aureus infections…” instead of “…S. aureus infections…”

L66: “…clinical forms of mastitis…” instead of “…clinical forms…”

L70: “…the SCC and the bacteriological…” instead of “…the SCC or the bacteriological…”

L73-75: SSC is an indicator of cells presence in milk including inflammatory cells. So, it’s an indicator of intramammary inflammation (mastitis) and in consequence predictor of intramammary infection. I suggest to report this difference.

L109-112: your main aim is to validate this new method. So, please make this explicit. Here you report information about M&M.

L129: there are any information regarding the quality control of this method?

L235-239: This paragraph should be placed in M&M

L244-246: This sentence should be placed in M&M

L246-249: All of these correlation are significant? Please report the p value for each one , including in figs. 1 to 5 (the graphic for non-significant values should be removed if the case).  Also add the legend.

L293 and L296: I suggest to remove Fig. 8 and 9. The values are not statistically different.

L256: What finding? The lack of significant and high correlation between S. aureus and SCC?

L257: “ … to identify intramammary infections caused by S. aureus…” instead of “ … to identify intramammary infections …”

L268: Table 3 is missing. These values are incorporated on Table 2.

L273: Please elucidate the reader about the independent method used to determine true and false positives/negatives

L299-312: I suggest to rewrite the conclusions according your main finding: A great sensibility  of isoleucine (Ile) and Leucine (Leu) was observed in this study suggesting they suitability as biomarkers for diagnosis of subclinical mastitis caused by S. aureus.

Author Response

(The authors gave the same response as above.)
